# An Intestinal Symbiotic Bacterial Strain of *Oscheius chongmingensis* Modulates Host Viability at Both Global and Post-Transcriptional Levels

**DOI:** 10.3390/ijms232314692

**Published:** 2022-11-24

**Authors:** Chengxiu Zhan, Long Chen, Dandan Guo, Jing Sun, Yunbin Duan, Panjie Zhang, Pengpeng Li, Lijun Ma, Man Xu, Ying Wang, Haoran Bao, Guofu Gao, Liwang Liu, Keyun Zhang

**Affiliations:** 1College of Life Sciences, Nanjing Agricultural University, Nanjing 210095, China; 2National Key Laboratory of Crop Genetics and Germplasm Enhancement, Key Laboratory of Biology and Germplasm Enhancement of Horticultural Crop (East China), Ministry of Agriculture and Rural Affairs, College of Horiticulture, Nanjing Agricultural University, Nanjing 210095, China

**Keywords:** entomopathogenic nematodes, symbiotic bacteria, transcriptome, miRNAs, molecular mechanism, interaction, viability, *rtk*, *fak*, miR-71, *daf-16*, microbiome, post-transcriptional level

## Abstract

A rhabditid entomopathogenic nematode (EPN), *Oscheius chongmingensis*, has a stable symbiotic relationship with the bacterial strain *Serratia nematodiphila* S1 harbored in its intestines and drastically reduced viability when associated with a non-native strain (186) of the same bacterial species. This nematode is thus a good model for understanding the molecular mechanisms and interactions involved between a nematode host and a member of its intestinal microbiome. Transcriptome analysis and RNA-seq data indicated that expression levels of the majority (8797, 87.59%) of mRNAs in the non-native combination of *O. chongmingensis* and *S. nematodiphila* 186 were downregulated compared with the native combination, including strain S1. Accordingly, 88.84% of the total uniq-sRNAs mapped in the *O. chongmingensis* transcriptome were specific between the two combinations. Six DEGs, including two transcription factors (*oc-daf-16* and *oc-goa-1*) and four kinases (*oc-pdk-1*, *oc-akt-1*, *oc-rtk*, and *oc-fak*), as well as an up-regulated micro-RNA, oc-miR-71, were found to demonstrate the regulatory mechanisms underlying diminished host viability induced by a non-native bacterial strain. *Oc-rtk* and *oc-fak* play key roles in the viability regulation of *O. chongmingensis* by positively mediating the expression of *oc-daf-16* to indirectly impact its longevity and stress tolerances and by negatively regulating the expression of *oc-goa-1* to affect the olfactory chemotaxis and fecundity. In response to the stress of invasion by the non-native strain, the expression of oc-miR-71 in the non-native combination was upregulated to downregulate the expression of its targeting oc*-pdk-1*, which might improve the localization and activation of the transcription factor DAF-16 in the nucleus to induce longevity extension and stress resistance enhancement to some extent. Our findings provide novel insight into comprehension of how nematodes deal with the stress of encountering novel potential bacterial symbionts at the physiological and molecular genetic levels and contribute to improved understanding of host–symbiont relationships generally.

## 1. Introduction

The intestinal microbiome is increasingly recognized as playing a major determining role in fundamental host physiology, affecting development and longevity, as well as in disease therapy [1,2,3,4,5,6,7]. The intestinal microbiome can fine-tune host longevity by increasing secretion of colanic acid through the host UPR^mt^-responsive transcription factor ATFS-1 in mitochondria [3] and can be modulated or activate metabolic drugs to treat cancer [2,4]. The gut microbiome also contributes to the development and function of the nervous system and to the balance between mental health and disease [5]; in mice, it determines levels of circulating cholesterol and may thus represent a novel therapeutic target in the management of dyslipidemia and cardiovascular diseases [6]. Unique host and microbiome features can achieve individualized gut mucosal colonization resistance to empirical probiotics [7]. However, an understanding of the molecular mechanisms involved in interactions between a member of the intestinal microbiome and the host remains elusive.

Entomopathogenic nematodes (EPNs) from the families Steinernematidae and Heterorhabditidae, as well as the genus *Oscheius* (*Heterorhabditidoides*) of the family Rhabditidae, have symbiotic relationships with single bacterial strains from the genera *Xenorhabdus*, *Photorhabdus*, or *Serratia* harbored in their intestines [8,9,10,11,12,13]. The single-strain intestinal bacterial partner of EPNs is essential for the pathogenicity of the nematode/bacterium complex and even for nematode development and reproduction [11,14,15,16,17,18]. The nematode host always achieves the best viability when its native bacterial symbiont is present [14,19,20]. As long as the bacterial strain is not toxic to the nematode host, some symbiotic bacteria can also provide nutritional support for nonspecific nematodes [21] and can even support the growth and reproduction of non-symbiotic nematodes [22,23]. Growth and reproduction of EPNs in the family Heterorhabditidae demonstrate a strict dependence on and specificity to their native symbiotic bacterial strain, whereas EPNs in families Steinernematidae and Rhabditidae have varying degrees of dependence on different bacterial symbionts and achieve stable or alternative rates of reproduction with a non-native symbiont [14,19,20,23,24,25]. Using axenic *Heterorhabditis* and different nonsymbiotic bacteria to form combinations, it was found that *Heterorhabditis* could not carry nonspecific symbiotic bacteria even at different stages of the host life cycle [26]. The most important aspects of the endosymbiotic relationships of EPNs and their associated bacteria are nutrition, specificity, pathogenicity, and viability of the inoculated nematode host. Some of these relationships are well characterized, and the nature of mutualism between the EPN and intestinal bacterial symbiont is well described [14,15,16,17,18,19,20,22,23,24,25,26]. However, a detailed understanding of the interaction between the EPN and its single-strain bacterial symbiont and the molecular mechanisms involved remains elusive.

The emergency of the era of big data provides a convenient way to comprehensively analyze the interaction mechanism between EPNs and their symbiotic bacteria in the perspective of the whole genome and transcriptomics. The genome and transcriptome analysis of four species of *Steinernema* and *Heterorhabditis bacteriophora* suggested that some unigenes in the transcriptome of EPNs might be related to their symbiotic interactions with symbiotic bacteria and parasitic insects [27,28,29]. A sRNA ArcZ in *Photorhabdus luminescens* was found to be involved in the regulation of the symbiotic relationship between the bacterium and the *Heterorhabditis* host by regulating its secondary metabolites [30]. These genomic and transcriptome studies provide new ideas/clues for the study of key molecules in EPNs responding to symbionts and the related molecular interaction mechanisms among those processes.

In contrast to the highly strain-specific EPNs of Heterorhabditidae, the EPN *Oscheius chongmingensis* in the family Rhabditidae, being mutually symbiotic with the bacterial strain *Serratia nematodiphila* DZ0503SBS1 (S1), has been shown to develop and reproduce harboring a non-native strain of *S. nematodiphila* DR186 (186) in this study. However, the viability of *O. chongmingensis* declines drastically when associated with the non-native strain 186. These characteristics make *O. chongmingensis* a good model for understanding the interactions and molecular mechanisms involved between a single member of the intestinal microbiome and the nematode host. In this study, we use the nematode *O. chongmingensis* cultured with both native and non-native strains of *S. nematodiphilia* to examine differences in gene expression and molecular genetic mechanisms underlying several aspects of host viability, including longevity, stress tolerance, and olfactory chemotaxis. Our findings reveal that a single intestinal bacterial species, *S. nematodiphila*, regulates these vital processes of *O. chongmingensis* at the transcriptional and post-transcriptional levels, and how the host responds to maintain a stable symbiotic relationship between them.

## 2. Results and Discussion

### 2.1. The Pathogenicity of Oscheius chongmingensis Is Correlated with Symbiotic Bacteria Carriage Rate and Declines Sharply with the Non-Native Strain Serratia Nematodiphila DR186 after Multigenerational Transfer

Initial experiments assessing nematode viability and symbiotic bacterial carriage ability were performed with four nematode/bacterium combinations: DZ/S1 (*Osheius chongmingensis* with its native *Serratia nematodiphila* S1), DZ/186 (*O. chongmingensis* with *O. rugaoensis’* native *S. nematodiphila* DR186 strain), and RG/186 and RG/S1 (*O. rugaoensis* with these same strains). After multigenerational transfer, the pathogenicity of IJs of the non-native combination DZ/186 against waxworms declined very rapidly; however, pathogenicity of the other three combinations of *Oscheius* and *S. nematodiphila* strains showed no significant decrease (Figure 1A–C and Appendix A). The lethal rates of IJs (infected juveniles) of DZ/186 against waxworms was the lowest and the most rapid decline at 72 h from 90% in the first generation to 25% in the fifth generation (Figure 1B,C and Appendix A), whereas that of the multi-generation IJs of the native combination DZ/S1 at 72 h remained stable with the same lethal rate (100%) as the first generation (Figure 1B). Symbiotic bacteria of EPNs are believed to play a leading role in insect host mortality; therefore, we tested changes in the bacterial carriage rate of the four monoxenic nematodes over multiple generations. The rate of strain-carrying IJs in all four nematode/bacterium combinations was high (78–95%) after the first infection cycle; however, after in vivo multigenerational transfer with waxworms, only the recombinant DZ/186 complex showed a sharply decreasing trend (from 81% to 10%) in terms of bacterial carriage rate (Figure 1D). The bacterial carriage rate of IJs of the other three combinations declined more slightly (Figure 1D). These results suggest that *O. chongmingensis* shows a strong specialization and stability for carrying its native symbiotic bacterial strain S1 compared with its very weak capacity for carrying the non-native strain; however, *O. rugaoensis* is not as strongly specialized toward its native strain 186 and has a greater ability to carry non-native strains of *S. nematodiphila*. These results indicated that *O. chongmingensis* can be a good model for understanding the molecular mechanisms and interactions involved between a nematode host and a member of its intestinal microbiome.

### 2.2. The Non-Native Bacterial Strain 186 Reduces the Fecundity of O. chongmingensis, Shortens Its Longevity, Impairs Its Stress Tolerance, and Decreases Its Olfactory Chemotaxis

The survival potential of *O. chongmingensis* declined markedly when associated with the non-native strain *S. nematodiphila* 186, even if it was of the same species as the native one. The developmental recovery rate of *O. chongmingensis* IJs was 85.43% with the native strain S1 and 26.72% with the non-native strain 186, whereas the developmental rate of *O. rugaoensis* was 75.19% with S1 and 83.52% with 186 (Table 1). The number of the second generation IJs of the recombinant DZ/186 complex per insect larva was the lowest (386), 16.5-fold less than that of the native DZ/S1 (6782) (Table 1). The average body size of *O. chongmingensis* associated with non-native 186 was also dramatically smaller than that with the native S1 (Table 1), which is similar to two *Steinernema* EPNs, *S. feltiae*, and *S. carpocapsae* fed with non-native bacterium [31].

The non-native bacterial strain 186 not only caused survival potential of *O. chongmingensis* declined but also significantly shortened its longevity (19 d) compared with the native strain S1 (21 d) in liquid nematode growth medium (NGM) (Figure 1E). The maximum longevity of the nematode was reduced from 24 days when associated with S1 to 22 days when associated with 186, and the median longevity was shortened from 14 (S1) to 10 (186) days accordingly (Table 2). The stress tolerance of *O. chongmingensis* was also positively correlated with longevity in the present study, similar to the situation with free-living nematodes [32]. We tested the heat and oxidative stress tolerances of 7 day-post L3 (third-stage larvae) of *O. chongmingensis*. The nematodes’ tolerance to both stresses in liquid NGM was markedly decreased associated with strain 186 (Figure 1F,G), and its median survival was one-third shorter (18 h) than that with strain S1 (24 h) for heat stress (Appendix A), which was 10.5 h with 186 and 15 h with S1, respectively, for oxidative stress tolerance (Appendix A).

Additionally, the non-native strain 186 also significantly decreased the olfactory chemotaxis response of *O. chongmingensis* to waxworms compared with that using the native strain S1 (Figure 1H).

### 2.3. Expression Levels of the Majority of Genes Involved in Biological Processes and Related Pathways in O. chongmingensis Associated with Strain 186 Are Downregulated Compared with Those in the Native Combination

The emergency of the era of big data provides a convenient way to comprehensively analyze the interaction mechanism between EPNs and their symbiotic bacteria from the perspective of the whole genome and transcriptomics. Using high-throughput sequencing, de novo assembly, and RNA-seq analysis, we obtained reliable digital gene expression (DGEs) of DZ/186 and DZ/S1 and detected 10,043 differentially expressed genes (DEGs) between the two monoxenic combinations (Figure 2A and Appendix A). Most of these DEGs (8797, 87.59%) were downregulated in the non-native combination (Figure 2A and Appendix A). Validation of DEGs was confirmed by detecting expression levels of ten randomly selected up- or downregulated unigenes using RT-qPCR analysis (Appendix A). Among downregulated DEGs, 4283 were assigned to 30 GO (gene ontology) groups belonging to three main categories, with 65.5% enriched with respect to biological processes, 22.8% for cell components, and 11.7% for molecular function (Appendix A). In total, 5439 DEGs were distributed among 255 KEGG pathways; the top 100 enriched pathways are shown in Figure 2B. Among these enriched pathways, exclusive of pathways with *Q*-value ≤ 0.05, focal adhesion (ko04510) was the most significantly enriched pathway, followed by ECM–receptor interaction (ko04512). In the focal adhesion pathway, 283 out of 312 DEGs were downregulated, indicating that genes involved in this pathway had a tendency toward reduced expression in DZ/186. Similar results were found for genes in all the other pathways, including regulation of actin cytoskeleton (ko04810), and pathways for longevity regulation (ko04212), insulin signaling (ko04910), MAPK signaling (ko04010), ErbB signaling (ko06042), Jak-STAT signaling (ko04630), and Toll-like receptor signaling (ko04620). These pathways are closely involved in nematodes’ longevity, olfactory responses, development, and reproduction [33,34,35].

According to the RNA-seq data, the majority of genes in the non-native combination showed significantly up- or downregulated expression, suggesting that the globally variable gene expression levels might result in the decreased viability of DZ/186. To further reveal how key DEGs that play vital roles in this reduced viability work, we selected *fak* and *rtk* in the focal adhesion signaling pathway, the most DEGs enriched pathway, and other four downregulated (*akt-1*, *daf-16*, *daf-36*, *fak*, *rtk* and *pdk-1*) and two upregulated (*goa-1* and *gsa-1*) DEGs involved in the top DEG-enriched signaling pathways related to longevity, olfaction, development, and reproduction for further functional and interaction mechanism identification in *O. chongmingensis* (Figure 2C and Appendix A).

### 2.4. Downregulated Oc-akt-1, Oc-fak, Oc-rtk, and Oc-pdk-1 as well as Three Transcription Factors, Oc-daf-16, Oc-goa-1, and Oc-gsa-1, Were Involved in the Regulatory Mechanism of Decreased Viability of O. chongmingensis

Two DEGs, *oc-daf-16* and *oc-fak*, were confirmed in our previous research to positively regulate longevity and stress resistance in *O. chongmingensis* [36,37]. The RNA interference (RNAi) technique was also employed to knock down expression of the other three genes, with 75.1%, 46.8%, and 25.7% lower expression levels for *oc-akt-1, oc-pdk-1*, and *oc-rtk* in RNAi *O. chongmingensis* than those in the corresponding control group (Figure 3A–C). Longevity and tolerance to heat stress were significantly prolonged in *oc-akt-1*:RNAi and *oc-pdk-1*:RNAi *O. chongmingensis* (Figure 3D,E,G,H,J,K). By contrast, longevity was reduced in *oc-rtk*:RNAi *O. chongmingensis* nematodes (Figure 3F,I). The *oc-pdk-1*:RNAi nematodes also gained increased oxidative stress tolerance (Figure 3L). These results revealed that downregulating the expression of *oc-daf-16, oc-fak*, and *oc-rtk* decreased longevity and stress tolerance, whereas downregulating *oc-pdk-1* and *oc-akt-1* induced longevity extension and stress resistance enhancement in *O. chongmingensis*.

*Oc-daf-36* was found to mediate the development process of *O. chongmingensis*. Despite the average body size of adults of *oc-daf-36:*RNAi *O. chongmingensis* being similar to that of L4440 nematodes, however, about 10% IJs of the former grew more slowly than the latter in 18 h, 30 h, and 42 h IJs-post (Figure 4A–C).

Two guanine nucleotide-binding proteins, *goa-1* and *gsa-1*, negatively regulate olfaction and fecundity in *C. elegans* [38,39]. The expression of *oc-goa-1* and *oc-gsa-1* in DZ/186 was upregulated seven-fold and a half-fold, respectively, compared with DZ/S1 (Appendix A) [39]; accordingly, both of the chemotaxis of DZ/186 decreased markedly compared with that of the native combination (Figure 1H) [40]. We considered whether *oc-goa-1* acts as an essential factor that negatively regulates the olfactory chemotaxis of *O. chongmingensis.* Therefore, we assessed the chemotaxis of *oc-goa-1*:RNAi *O. chongmingensis* to waxworms and benzaldehyde, respectively (Figure 4D–F). As expected, the chemotaxis of *oc-goa-1*:RNAi nematodes increased one- to two-fold more than that of nematodes treated with the blank control L4440 (Figure 4E,F). The amount of eggs laid was also increased in *oc-goa-1*:RNAi *O. chongmingensis* compared with the L4440 control (Figure 4G), further demonstrating that *oc-goa-1* also negatively regulates fecundity in *O. chongmingensis. Oc-rtk*, as well as *oc-fak* [37], was found to positively mediate the olfaction and fecundity of *O. chongmingensis* (Figure 4H,I). The chemotaxis of *oc-rtk*:RNAi nematodes decreased 2.5-fold more than that of nematodes treated with the blank control L4440 (Figure 4H). The average body size of *oc-rtk*:RNAi *O. chongmingensis* was dramatically smaller than that of L4440 nematodes (Figure 4I).

Among the eight selected DEGs, two tyrosine kinases, *oc-fak* and *oc-rtk*, were found to be involved in the regulation of various aspects of the viability of *O. chongmingensis*, including longevity, fecundity, and olfaction (Figure 3F,I and Figure 4H,I) [37]. *Oc-fak* and *oc-rtk* not only positively mediated the expression of each other but also mediated that of *oc-daf-16* (Figure 5A,B), which demonstrates that *oc-fak* and *oc-rtk* impact longevity by mediating the expression of *oc-daf-16*. The expression level of *oc-goa-1* increased by 62.3% and 33.3%, respectively, in *oc-fak*:RNAi *O. chongmingensis* and in *oc-rtk*:RNAi *O. chongmingensis*, which reveals that *oc-fak* and *oc-rtk* also influence olfaction by negatively regulating the expression of *oc-goa-1*. These results confirmed that both *oc-fak* and *oc-rtk* play key roles in the viability regulatory mechanism of *O. chongmingensis* by positively mediating the expression levels of each other and *oc-daf-16*, and by negatively mediating that of *oc-goa-1*.

### 2.5. High-Throughput Sequencing Indicates More Than 88.84% MicroRNAs in the Two Combinations Were Significantly Differentially Expressed

MicroRNAs (miRNAs) comprise a large family of small (~22-nucleotide-long) noncoding RNAs that have emerged as key post-transcriptional regulators of gene expression in plants, metazoans, and mammals [41,42,43,44]. In mammals, miRNAs have been predicted to regulate the activity of more than 60% of all protein-coding genes [45]. To determine whether and how miRNAs of *O. chongmingensis* are related to the mechanism regulating expression of DEGs involved in decreased viability of the non-native combination, we used next-generation sequencing technology to identify differentially expressed miRNAs between the native and non-native combinations across the genome. The majority of the small RNAs (sRNAs) of the two combinations were 21 nucleotides (nts) in length (Appendix A). By mapping sRNA libraries to the reference sequences of the *O. chongmingensis* transcriptome, we found that 88.84% of unique sRNAs were specific between DZ/S1 and DZ/186 (Figure 6A); of the rest, 179 differentially expressed miRNAs were identified, 123 (68.49%) of which tended to be upregulated in the non-native combination (Figure 6B). The relative expression levels of the random selected six upregulated and four downregulated miRNAs were consistent with sequencing data (Appendix A). Further analysis using GO function prediction demonstrated that these miRNAs were mostly enriched with respect to biological processes (Appendix A). Among these differentially expressed miRNAs, 29 known and 5 novel miRNA families showed upregulated patterns, and another 5 miRNAs showed downregulated patterns in libraries of the non-native combination (Appendix A).

### 2.6. Oc-miR-71 Mediates the Longevity and Stress Tolerance of O. chongmingensis by Targeting Oc-pdk-1

We predicted miRNA–mRNA interactions for the aforementioned eight DEGs with verified functions involved in the regulatory mechanism of decreased viability of DZ/186 using conjoined analysis of DEGs and sRNA libraries. A dual-luciferase reporter assay was employed to verify the interaction in vitro between miRNAs and their potential targets except for those (*oc-akt-1*, *oc-daf-16,* and *oc-daf-36*) whose interaction sites were not in the 3′ UTR or those with low interaction prediction scores (Figure 6C). Precursor sequences of those miRNAs were not found in the high-throughput sequencing database of *O. chongmingensis*; hence, we used mimic miRNA, a chemical synthetic RNA single-stranded RNA miRNA function identified in vitro (dual-luciferase reporter assay) and in vivo. Four genes, *oc-fak, oc-goa-1*, *oc-gsa-1*, *and oc-rtk*, were not targets of the predicted miRNAs (Figure 6D–G), whereas *oc-pdk-1* was confirmed to be a target of oc-miR-71 in vitro (Figure 6H). The targeting relationship between oc-miR-71and *oc-pdk-1* in *O. chongmingensis* needs to be further verified in vivo.

The expression of oc-miR-71 was significantly increased in DZ/186 compared with DZ/S1 (Figure 7A), and also in *O. chongmingensis* stressed with heat at 37 °C or with paraquat (10 mM) for 5 h (Figure 7B,C). All these results confirm that oc-miR-71 is positively related to stress tolerance exhibited by *O. chongmingensis*. The sequence of oc-miR-71 has 100% identity with miR-71 of *C. elegans*, and miR-71 also plays regulatory roles in the longevity and stress tolerance of *C. elegans* [46,47]. As mentioned previously, *oc-pdk-1,* the verified target of oc-miR-71 in vitro, is negatively correlated with longevity and stress tolerance in *O. chongmingensis* (Figure 3E,H,K,L and Figure 6H). We employed tests in vivo to further verify the targeting relationship between oc-miR-71and *oc-pdk-1*. Overexpression of oc-miR-71 was detected after soaking *O. chongmingensis* IJs in mimic oc-miR-71 for 12 h (Figure 7D), and as expected, the expression of *oc-pdk-1* was significantly reduced accordingly (Figure 7E), and both longevity and stress tolerance of the treated nematodes decreased significantly (Figure 7F,G). These results confirmed that oc-miR-71 regulates longevity and stress tolerance of *O. chongmingensis* at least in part by negatively regulating the expression of *oc-pdk-1*.

### 2.7. The Non-Native Intestinal Bacterial Strain 186 Mediates O. Chongmingensis’ Viability and Symbiotic Regulatory Response at Both Global and Post-Transcriptional Levels

The intestinal microbiome plays a crucial role in mediating diverse host biological processes, including host–symbiont interactions [1,2,3,6]. The present study offers a model for understanding the symbiotic interactions between a nematode host and a single intestinal bacterial strain at the beginning of the symbiotic evolution. *Oscheius chongmingensis* has a stable mutually symbiotic relationship with its native bacterial partner *Serratia nematodiphila* S1 and cannot develop without the strain being present [11]. In the present study, the non-native bacterial strain 186, which is of the same species as the native one, proved to be a stress to the survival of *O. chongmingensis* and resulted in decreased nematode viability. High-throughput sequencing indicated that in comparison with the native combination, global expression levels of mRNAs in the non-native combination of *O. chongmingensis* and *S. nematodiphila* 186 exhibited downregulated trends (87.59%); accordingly, 88.84% of the unique sRNAs of the two libraries were different, and 68.49% of the remained miRNAs tended to be upregulated in the non-native combination. Further research demonstrated that of these DEGs and differentially expressed miRNAs, there were five downregulated genes, two upregulated genes, and one miRNA, the upregulated miR-71, all of which were confirmed to be involved in the regulating mechanism of viability and stress responses of *O. chongmingensis* associated with the non-native strain 186 (Figure 8). Our findings demonstrated that the single intestinal bacterial species *S. nematodiphila* orchestrates its nematode host’s viability at the global transcriptional and post-transcriptional levels, in part by regulating *oc*-*rtk*, *oc-fak*, *oc*-*akt-1*, and oc-miR-71 (Figure 8).

In the predicted regulating network (Figure 8), *oc-fak* and *oc-rtk* positively mediated olfaction and fecundity in *O. chongmingensis* in transcriptional levels by mediating the expression level of *oc-goa-1* (Figure 4H,I and Figure 5A,B) [37]. In the non-native combination, RNAi knockdown *oc-fak* and *oc-rtk* both led to the increased expression of *oc-goa-1*. *Oc-goa-1* was confirmed to negatively regulate the fecundity and olfactory capabilities of *O. chongmingensis* in this study (Figure 4E–G). Thus, we deduced that the downregulation of *oc-fak* and *oc-rtk* bring about the upregulated expression of *oc-goa-1* and then indirectly reduce the fecundity and olfactory capabilities of *O. chongmingensis* (Figure 8). This is the first report showing that both of the two kinds of tyrosine kinases, the non-receptor FAK and a member of receptor tyrosine kinases (RTKs), mediate positively olfactory chemotaxis of nematodes by negatively regulating the expression of *goa-1*. Previous studies only found that some members of RTKs and *goa-1* counteract each other in the regulation of the regeneration of axons in *C. elegans* [48,49]. Further study needs to reveal the related regulatory mechanism among the three genes.

The gene *daf-16*, encoding a single FOXO/DAF-16 transcription factor, is a central regulator of multiple signaling pathways that receives and integrates signals to regulate several biological processes, including longevity, development, reproduction, fat storage, stress tolerance, and innate immunity [33,36,46,47,50,51,52]. Genetic and molecular analysis showed that *daf-16* is a downstream conservative target of the ILS pathway. The PI3 kinase/AKT signaling cascade ultimately controls the nuclear localization of DAF-16 [50]. FOXO/DAF-16 was identified as a central regulator of the aging process in *C. elegans* [51]. In *C. elegans*, the ILS pathway acts both cell-autonomously and non-autonomously to control longevity through regulation of the nuclear localization and transcriptional activation of FOXO/DAF-16 [33,52]. In this study, we found that the expression level of *oc-daf-16* in the non-native combination was significantly lower than in the native combination, prompting speculation that the downregulation of *daf-16* contributes to shorter longevity and weaker stress tolerance in the non-native combination. Interestedly, we found that both downregulated *oc-fak* and *oc-rtk* caused the downregulated expression level of *daf-16* in *oc-fak* and *oc-rtk* RNAi *O. chongmingensis*, which exhibited both shorter longevity and weaker stress tolerance.

*Drosophila* mutants lacking Fak56 also exhibited a decreased lifespan [53]. Fifty-one proteins in the focal adhesion signaling pathway are involved in or at least associated with the control of longevity [54]. Along with the insulin pathway, the common signaling network is remarkably enriched with the focal adhesion and adherens junction proteins, whose relation to the control of lifespan is yet to be fully addressed [54]. In the present study, we found that RNAi knockdown *oc-fak* resulted in the downregulating expression of daf-16, which demonstrated that *oc-fak* positively mediates the longevity of *O. chongmingensis* by positively regulating the expression of *oc-daf-16.* The overexpression of *tkr-1(old-1)*, encoding a member of RTKs, increased longevity 40–100% in transgenic *C. elegans* [55], and further study revealed that *old-1* expression is dependent on *daf-16* [56]. However, in the present study we found that *oc-rtk* positively mediates the longevity of *O. chongmingensis* by positively regulating the expression of *oc-daf-16.* These results suggested that the reduced expression levels of *oc-fak* and *oc-rtk* in the non-native combination resulted in shortened longevity and reduced stress-resistance of the nematode by regulating the expression of *oc-daf-16* (Figure 8).

We also deduced a response mechanism model at both transcriptional and post-transcriptional levels for *O. chongmingensis* to enhance its survival when faced with a non-native intestinal bacterial strain (Figure 8). Survival is crucial for organisms when faced with stress. Although *O. chongmingensis* could grow on plates with a lawn of the non-native bacterial strain 186, which is of the same bacterial species as the native S1 strain, its viability declined very rapidly over several generations. This result suggested that colonization by the non-native bacterial strain 186 represents a stress to the survival of the nematode. The decreases in longevity and stress tolerance represent a threat to the survival of the organism, which can induce *O. chongmingensis* to make adjustments in its response to the stress of invasion by the non-native bacterial strain 186. At the transcriptional level, expression of *oc-akt-1* is repressed in the non-native combination, which will reduce the phosphorylation level of DAF-16 and result in more DAF-16 being imported into the nucleus to enhance longevity and tolerance to high-temperature and oxidative stresses via the insulin-like signaling pathway [50,52]. At the post-transcriptional level, the upregulation of oc-miR-71 downregulates the expression level of the target gene *oc-pdk-1* in the ILS pathway to decrease the phosphorylation level of oc*-*AKT-1, improving the longevity and stress tolerance of *O. chongmingensis* [50,52] (Figure 8).

## 3. Materials and Methods

### 3.1. Nematodes and Symbiotic Bacterial Strains

*O. chongmingensis* DZ0503CMFT (DZ), *O. rugaoensis* RG081015 (RG), and their symbiotic bacterial strains *S. nematodiphila* DZ0503SBS1 (S1) and DR186 (186) were identified by and maintained in our laboratory [11,12,57]. The nematodes used in this study were raised on waxworms (fifth-instar larvae of the wax moth *Galleria mellonella*) or raised on lipid agar (LA, containing 8 g nutritional broth, 5 g yeast extract, 2 g MgCl2·6H_2_O, 4 mL corn oil, 7 mL corn syrup, 1 g sodium pyruvate, and 15 g agar in 1 L culture medium) plates with a bacterial lawn of *S. nematodiphila* [22,58], on liver/agar (CA) plates [24], and on liquid NGM [59].

### 3.2. Axenic IJs and Combinations of Monoxenic Nematodes with Native and Non-Native Bacterial Strains

Axenic eggs of each nematode species were obtained by crushing 40 mature females in sterile Ringer’s solution (containing 100 mM NaCl, 5 mM HEPES, 2 mM CaCl_2_, 1.8 mM KCl, and 1 mM MgCl_2_, pH 6.9) with sodium hypochlorite (2.5%, *w/v*) and 0.4 M NaOH for 10 min. Released eggs were rinsed three times with sterile Ringer’s solution and collected by centrifugation. A portion of those axenic eggs were used to obtain axenic IJs according to methods by Sicard et al. [24]. Axenic eggs were inoculated onto separate CA plates and cultured at 25 °C for about three weeks until the IJs emerged. Four monoxenic nematode/bacterium combinations (DZ/S1, DZ/186, RG/186, and RG/S1) were prepared by a method modified from previous studies [14,24,58]. Specifically, axenic eggs were transferred to liver/agar plates and cultivated for 48 h at 25 °C to obtain second-instar larvae that were then transferred to LA plates with 2-day-old bacterial lawns of S1 or 186, respectively, and cultivated at 25 °C until the IJs emerged.

### 3.3. Pathogenicity of IJs from Different Infection Cycles of the Four Monoxenic Combinations against Waxworms

IJs of the four monoxenic combinations inoculated on the bacterial lawn were considered first-generation IJs. A total of 320 first-generation IJs was used to infest waxworms, and IJs emerging from the cadavers represented the second-generation IJs. This procedure was repeated to prepare the third- to fifth-generation IJs. We carried out experiments to test the pathogenicity of IJs against the insect host by using a modified previous method [60]. A total of 320 IJs in 0.9 mL of sterilized tap water was added to 9 cm petri dishes without filter paper. Ten waxworms were placed in each dish and cultivated at 25 °C with the same volume of sterile tap water and axenic IJs used as controls. Each treatment was replicated three times. The mortality of treated insects was recorded every 12 h, and the data were analyzed by one-way ANOVA followed by multiple pairwise comparisons using Tukey’s HSD (α = 0.05). All raw data passing these tests were applied to estimate the larval mortality means varying across multi-generation IJs, using GraphPad Prism 7.00 (GraphPad Software, San Diego, CA, USA). All other data in this study were statistically analyzed by ANOVA.

### 3.4. Bacterial Carriage Rate of the Four Nematode–Bacterium Combinations

IJs of each generation were surface-disinfected by incubating them in 0.12% sodium hypochlorite solution for 5 min, washing them three times with sterile Ringer’s solution, and harvesting them by centrifugation [60]. Each of 100 surface-disinfected IJs was crushed and subsequently streaked on an NBTA plate and cultivated at 30 °C for 24–48 h [8]. Bacterial colonies were subsequently counted to calculate the bacterial carriage rate (the number of IJs with bacteria-colonized intestines per 100 IJs).

### 3.5. Influence of S. Nematodiphila Strains on the Development and Reproduction of Two Species of Oscheius

The *S. nematodiphila* strains S1 and 186 were each streak-spread separately on LA medium and then cultivated at 30 °C for 48 h. One hundred monoxenic IJs of each of the four previously described combinations were put into cultures of each of the two bacterial strains, and the cultivation temperature was changed to 25 °C. After the primary adults emerged, the total number and percentage of females for all adult nematodes of each combination were counted to calculate the developmental rate (percentage of total IJs developed into adults) and female ratio. The number of subsequent IJs was counted after 15 days. From each combination, 40 subsequent IJs were randomly selected, killed in 60–65 °C water, and placed on temporary slides for measurement of body length and width under the microscope. Each treatment was replicated three times.

### 3.6. Longevity and Stress Tolerance Analysis

Longevity and stress tolerance assays for DZ nematodes were performed according to previously described methods with modifications [61]. For producing relatively synchronous populations of young adults for longevity and stress tolerance assays, IJs of the DZ nematodes were cultured with the appropriate bacterial strain in liquid NGM at 25 °C until the first few eggs were laid. Adults were subsequently extracted, and the eggs remained in the NGM until they reached the young adult stage. For the longevity assay, each young adult was transferred to a separate well on a 24-well plate containing liquid NGM with the appropriate freshly-cultured bacterial strain and monitored once daily until death. The young adult stage was defined as day 0 in the longevity assays. During the test, nematodes were transferred onto fresh 24-well liquid plates with cultured bacterial cells every 2 days and were scored as dead when they no longer responded phototactically or when the head was prodded. Longevity assays were repeated three times (n = 30 for each replicate). Age-related changes of nematode physiological processes, including reproductive span, fast body movement span, fast pharyngeal pumping span, and pharyngeal pumping, were calculated as per a previously described method with modifications [62]. Each well of a 24-well plate containing liquid NGM was placed with one infective juvenile, except for two IJs onto each well for a reproductive span assay. Each nematode was examined every day for the following phenotypes: (i) Reproduction was assessed by the presence of progeny; (ii) body movement was assessed by observation for 10 s. Nematodes are defined as having fast movement if they exhibit continuous and well-coordinated movement with a locomotion rate of >1 mm per 10 s; (iii) pharyngeal pumping was assessed by observing the number of pharyngeal contractions during a 10 s interval. Nematodes that displayed 0, 1–24, and >25 pharyngeal contractions per 10 s were considered as no pumping, slow pumping, and fast pumping, respectively; (iv) lifespan was defined from the L3 stage to the last day of survival. All indicators were observed from the same nematode. The experiments were implemented three times (n = 30 for each).

For the heat stress assay, each young adult was transferred to a separate well on a 24-well plate with sterile Ringer’s solution at 37 °C, and surviving nematodes were recorded every 3 h. For the paraquat assay, each young adult was exposed to 10 mM paraquat at 25 °C, and survival was scored every 3 h. Both the heat and paraquat assays were carried out three times (n = 30 for each).

### 3.7. Determination of Olfactory Chemotaxis by Nematodes

Chemotaxis of DZ populations was measured by establishing a gradient of attractant from a point source and observing the accumulation of nematodes at the attractant source using a modified previously described method [63]. A thin layer of agar in a 9 cm petri dish was used as a matrix for chemotaxis. About 200 IJs were placed in the center area of the plate, with the attractant at one end and a control counter-attractant at the opposite end. After 1 h, the number of IJs at the attractant area and the control area was counted. The attractant samples used were either benzaldehyde or odor from live waxworms. Absolute ethanol diluted to the same concentration as the benzaldehyde was used as the control for the benzaldehyde assay, while air was used as the control for the waxworm odor assay. A chemotaxis index was calculated based on the enrichment of IJs at the attractant and at the control as follows: chemotaxis index = (*n_A_* − *n_C_*)/(*n_A_* + *n_C_*), where *n_A_* is the number of nematodes within the attractant sphere, and *n_C_* is the number of nematodes within the control sphere [22]. The index could vary from +1.0 to −1.0.

### 3.8. Screening for Differentially Expressed Genes and Functional Classification

Total RNA was isolated separately from nematodes of combinations DZ/S1 and DZ/186 raised on waxworms for 18 days using TRIzol reagent (Invitrogen, Carlsbad, CA, USA), genomic DNA contamination was eliminated according to the manufacturer’s instructions, and subsequently total RNA was quantified using an Agilent 2100 assay. The total RNA, with m(RNA) ≥ 20 μg and RNA (integrality number) ≥ 8.5, was used for both digital gene expression (DGE) and sRNA library construction. cDNA libraries were constructed using previously described protocols [64] and subjected to deep sequencing using Illumina HiSeq^TM^ 2000 (Illumina, San Diego, CA, USA) at the Beijing Genomics Institute (BGI, Shenzhen, China). Only clean reads were selected for mapping to the reference transcriptome sequence set using SOAP2 [65], with no more than one mismatch allowed in the alignment. Clean reads that were mapped to reference sequences from multiple genes were filtered out; the remaining reads were designated as unambiguous reads. The gene expression level of unambiguous reads was calculated using the RPKM (reads per kilobase per million reads) method [66]. A rigorous algorithm was developed to identify the DEGs between two different DGE libraries using a previously described method [67]. We determined the threshold *p*-value in multiple tests and analyses by manipulating the false discovery rate (FDR) value. In the present study, FDR ≤ 0.001 and |log_2_Ratio| ≥ 1 were used as thresholds to screen for significant DEGs. The DEGs were assigned to gene ontology (GO) classifications using BLAST2GO software [68,69]. KEGG pathway annotation was performed against the KEGG database by Pathfinder software [70]. Pathways with a *Q*-value ≤ 0.05 indicated significant enrichment in DEGs.

### 3.9. RT-qPCR Validation of DEG Expression Levels

Random reverse transcription quantitative PCR (RT-qPCR) validation of the expression level of mRNAs tested in related nematodes was carried out as follows. Total RNA was extracted by using TRIzol as the aforementioned, with genomic DNA elimination. SuperQuick RT MasterMix (CWBIO, Beijing, China) was used to perform reverse transcription and to synthesize first-strand cDNA. Relative expression levels of ten randomly selected DEGs and eight DEGs of interest were determined to verify the DGE data in this study. All the primers used in this study were designed by Primer Premier 5.0, except for those of miRNAs, which were designed by tail-adding methods (Appendix A). qPCR was performed on the ABI Prism 7300 DNA sequence detection system (Applied Biosystems, Waltham, MA, USA) with a 20 µL fast real-time PCR system consisting of 2 µL of diluted cDNA, 0.4 µM forward and reverse primers, 0.4 µL of ROX Reference Dye (50f), and 10 µL of 2X SYBR Green PCR Master Mix (Takara Bio, Dalian, China) by the amplification program of incubation at 95 °C for 30 s, followed by 40 cycles at 95 °C for 15 s, and at 60 °C for 30 s. RT-qPCR reactions were performed with three technical repeats and three independent biological replicates. The relative quantitative method (*ΔΔC*^t^) was used to calculate the fold change of target genes, normalized to *18SrRNA* and relative to control expression [71]. Linear regression analysis of gene expression levels between DGE and RT-qPCR data and calculation of correlation coefficients (*r*) were performed with SPSS v.24.0 (IBM, Armonk, NY, USA).

### 3.10. SRNA Library Construction and Differential Expression Analysis of MiRNAs in DZ/S1 and DZ/186

The total RNAs of DZ/S1 and DZ/186 used for DGE library construction were simultaneously used for the construction of sRNA libraries. Two sRNA libraries of DZ/S1 and DZ/186 were prepared with previously described procedures [63]. Briefly, total RNA was separated and purified by 15% polyacrylamide gel electrophoresis. sRNAs of 18–30 nts were collected and ligated with 5′- and 3′-RNA adapters using T4 RNA ligase (Takara Bio). The adapter-ligated sRNAs were subsequently reverse-transcribed to complementary DNA (cDNA) using Super-Script II reverse transcriptase (Invitrogen, China) and then amplified by PCR. Finally, sRNA libraries were sequenced at the Beijing Genomics Institute (BGI-Shenzhen) using Solexa sequencing technology (Illumina, San Diego, CA, USA). Only clean reads were mapped to the nematode reference sequences containing genomic survey sequences, expressed sequence tag sequences, and mRNA transcriptome sequences using the program SOAP2 (BGI). sRNA sequences matching rRNA, tRNA, small nuclear RNA (snRNA), and small nucleolar RNA (snoRNA), as well as sequences containing poly(A) tails were excluded. The remaining sRNAs were analyzed by BLAST searches against animal miRNA mature sequences and precursor structures that were annotated in miRBase 20.0 to identify known miRNAs and base mutation miRNAs in the libraries. Novel miRNAs were predicted from the remaining unknown reads using MiReap software (https://sourceforge.net/projects/mireap/, accessed on 15 March 2014). The stem–loop structures of pre-miRNAs were estimated with Mfold [72]. Relative changes in expression levels of known miRNAs between the two libraries were assessed visually with a scatterplot. Fold change was calculated as log_10_(miRNA normalized reads in DZ/186/miRNA normalized reads in DZ/S1). *P*-values were calculated based on the aforementioned methods. The miRNA fold change was >1 or <−1, with *p*-values < 0.05 representing a significant difference in expression. RT-qPCR validation of miRNA expression levels was run and analyzed as described previously except for several special steps as follows. Tailing and reverse transcription of sRNA were performed simultaneously using a One Step PrimeScript^®^ miRNA cDNA Synthesis Kit (Takara Bio) with 20 µL of reaction mixture including 10 µL 2X miRNA Reaction Buffer Mix, 2 µL 0.1% bovine serum albumin, 2 µL miRNA PrimeScript^®^ RT Enzyme Mix, and 2 μL total sRNAs incubated at 37 °C for 60 min and at 85 °C for 5 s in a SensoQuest Labcycler PCR instrument (SensoQuest, Göttingen, Germany). The expression level of *5.8SrRNA* rather than that of *18SrRNA* was used as the reference to calculate the fold change of miRNAs in RT-qPCR.

### 3.11. Bioinformatic Prediction and Annotation of Potential Targets of Differentially Expressed MiRNAs

The potential targets of differentially expressed miRNAs were predicted using PicTar (https://pictar.mdc-berlin.de/, accessed on 18 March 2017) and RNAhybrid software [73]. Taking the intersection of the two predictions, we obtained the optimal potential target. For miRNAs for which no potential target was detected using either aforementioned software method, potential targets were predicted individually. To clarify the potential functions of the miRNA-targeted genes, GO and KEGG pathway analysis annotations were assigned using the DAVID gene annotation tool [74].

### 3.12. Luciferase Reporter Assay Validation of In Vitro MiRNA Targets

Validation of the in vitro interactions between miRNAs and their potential targets was performed according to a dual-luciferase reporter assay modified from previously described methods [75]. The synthetic RNA duplexes/single-stranded RNAs, mimic-miRNAs, and mimic nucleic acid control (mimic-NC) in this study were synthesized (RiboBio, Guangzhou, China) with identity to the sequences of related miRNAs, being developed for functional studies in vitro and in IJs of DZ nematodes, respectively. HEK293T cells (provided by Dr. Yan Zeng, Nanjing Agricultural University, Nanjing, China) were cultured in Gibco^TM^ Dulbecco’s modified Eagle medium (Thermo Fisher Scientific, Waltham, MA, USA) supplemented with 10% fetal bovine serum in 24-well plates and were co-transfected with pMIR-REPORT^TM^ luciferase vector (Applied Biosystems) containing either wildtype or mutated putative miRNA target sequences (Appendix A; designed according to the protocol of pMIR-REPORT^TM^) and mimic-miRNAs or mimic-NC using Lipofectamine 2000 (Invitrogen) according to the manufacturer’s instructions. Luciferase activities were measured at 48 h post-transfection with a dual-luciferase reporter assay kit (Promega, Madison, WI, USA) on a GloMax^®^ 20/20 luminometer (Promega), and *Renilla* luciferase activity was normalized to firefly luciferase activity.

### 3.13. Transfection of Mimic MiRNA into Nematode and Function Validation In Vivo

Nematode IJs were exposed to mimic-miRNAs (1 µM) and mimic-NC (1 µM) for 12 h, then collected and cultured individually in NGM in one well of a 24-well plate for 5–7 days. At 2-day intervals, they were retreated with mimic-miR-71 (1 µM) for 4 h to continuously transfect miRNA into nematodes [76]. About 150 nematodes cultured for 5–7 days in liquid NGM were collected immediately and frozen at −80 °C for RNA extraction. Expression levels of miR-71 and target gene in nematodes exposed to corresponding mimic-miRNAs were detected by RT-qPCR according to the aforementioned methods. In addition, the biological characteristics of each group of treated nematodes including body length, longevity, egg production, female ratio, chemotaxis, heat shock tolerance, and antioxidant capacity were recorded and analyzed using the aforementioned methods.

### 3.14. RNAi by Feeding

Recombinant fragments of the conserved regions of cDNAs of the miRNA targets and other functional genes were obtained as described previously and double-digested to produce suitable cohesive ends, then connected to the double-digested dsRNA expression vector L4440. The constructed plasmids were then transformed into *E. coli* HT115 (DE3) RNaseIII-deficient cells. As described previously [77], synchronized L1-stage nematodes were transferred to NGM containing seeded bacteria expressing dsRNA of relevant genes induced by 0.1 mM isopropyl-β-d-thiogalactopyranoside for 4–8 h. These P_0_ nematodes continued to be incubated at 20 °C until IJs emerged. These P_1_ IJs were then collected, and expression levels of the relevant genes were detected by RT-qPCR. Body length, longevity, female ratio, and other biological characteristics (except for eggs laid and developmental defects in each group of treated nematodes) were recorded and statistically analyzed as described previously. Eggs laid were recorded when nematodes began to oviposit, and the spawning amount of each group was recorded and counted for 48 h.

## 4. Conclusions

In summary, we investigated the reduced viability and pathogenicity of the entomopathogenic nematode *Oscheius chongmingensis* when co-cultured with a non-native strain of the symbiotic bacterium *Serratia nematodiphilia*. Based on digital gene expression profiles, we selected eight differentially expressed genes related to the most clearly affected aspects of viability. Using RNAi, we systematically evaluated these genes’ effects on host viability, demonstrating their involvement in molecular regulatory mechanisms of reduced host viability. We also identified a level of post-transcriptional regulation by a previously known miRNA to mediate longevity and stress tolerance through targeting of one gene in the ILS pathway. Our work demonstrates that the single intestinal bacterial species *S. nematodiphila* orchestrates its nematode host’s viability at the global transcriptional and post-transcriptional levels, in part by regulating *oc*-*rtk*, *oc-fak*, *oc*-*akt-1*, and oc*-*miR-71.

## Figures and Tables

**Figure 1 ijms-23-14692-f001:**
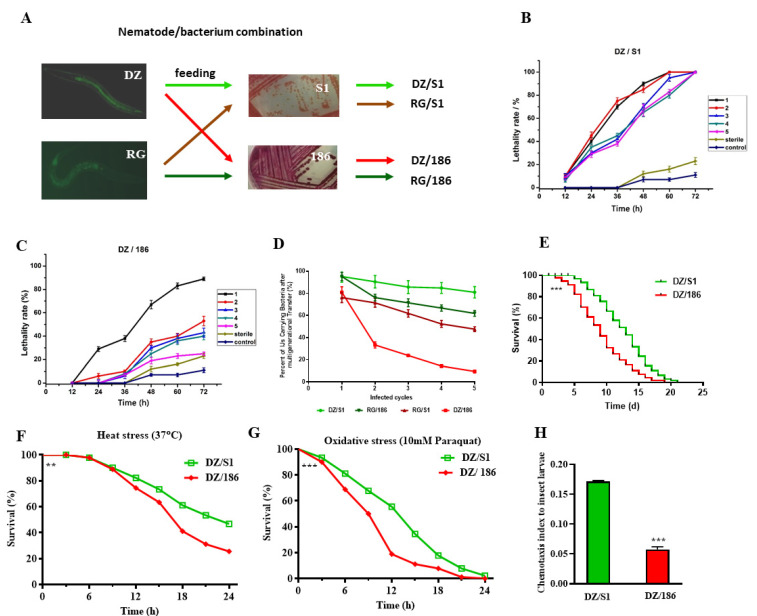
Symbiotic bacterial strains impact pathogenicity, longevity, stress resistance, and chemotaxis of *Osheius chongmingensis*. (**A**) The four monoxenic nematode/bacterium combinations of *O. chongmingensis* and *O. rugaoensis* associated, respectively, with *S. nematodiphila* S1 (DZ/186 and DZ/S1) and *S. nematodiphila* DR186 (RG/186 and RG/S1). (**B**,**C**) Pathogenicity changes of IJs of DZ/S1 and DZ/186, respectively, after multigenerational passage through *G. mellonella* in vivo. (**D**) Percent change in bacterial carriage of IJs of the four combinations after multigenerational passage. (**E**–**H**) Effects of symbiotic bacterial strains S1 and 186 on longevity, stress tolerance, and chemotaxis to benzaldehyde of *O. chongmingensis.* ** *p* < 0.01; *** *p* < 0.001.

**Figure 2 ijms-23-14692-f002:**
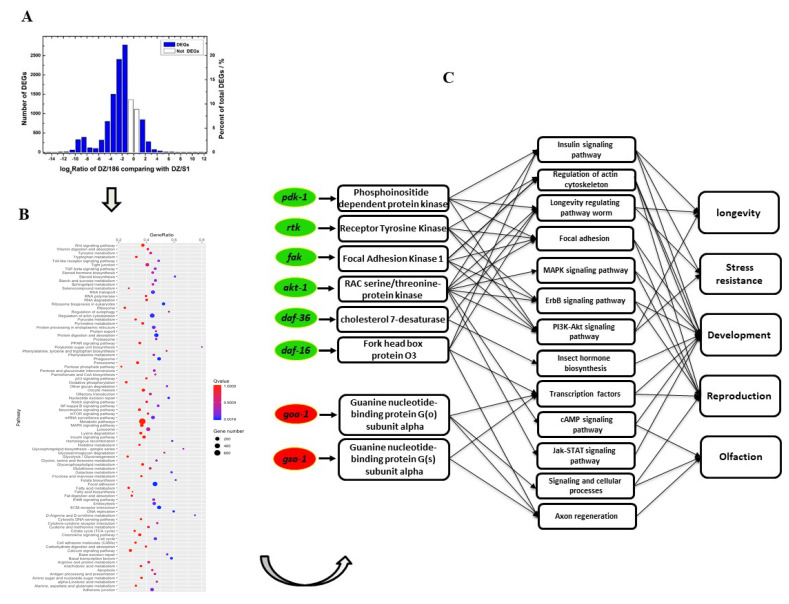
(**A**) Global comparison analysis of expression level of all mRNA data of native vs. non-native monoxenic *O. chongmingensi*. (**B**) The top 100 signaling pathways enriched in DEGs annotated by KEGG analysis. (**C**) Predicted functions of six downregulated and two upregulated DEGs in DZ/186 involved in regulating longevity, stress resistance, and chemotaxis, according to previous studies.

**Figure 3 ijms-23-14692-f003:**
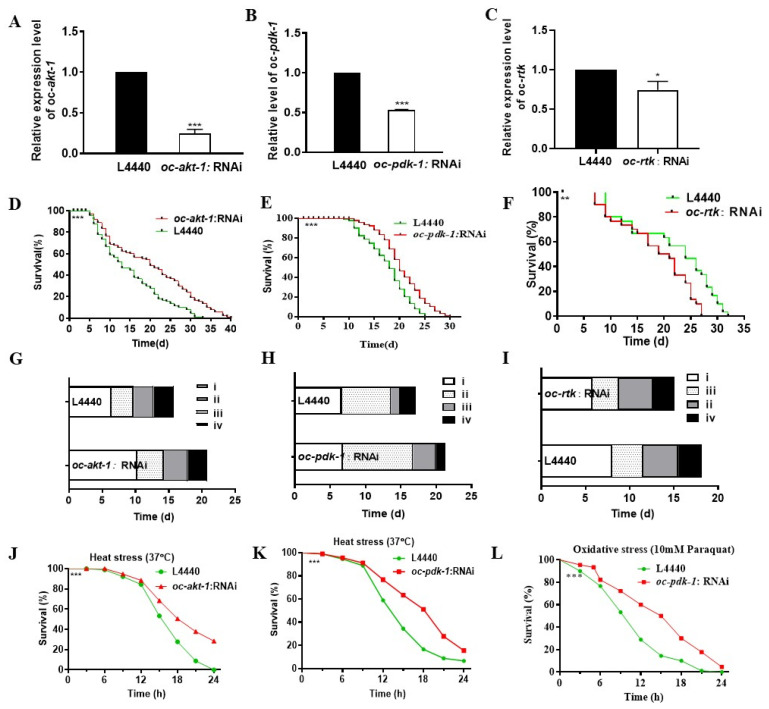
Identification of functions of *oc-akt-1*, *oc-pdk-1*, and *oc-rtk* in regulating longevity and stress resistance of *O. chongmingensis*. (**A**) The expression level of *oc-akt-1* is knocked down by 75.1% in *oc-akt-1*:RNAi nematodes. (**B**) The expression level of *oc-pdk-1* is knocked down by 46.8% in *oc-pdk-1*:RNAi nematodes. (**C**) The expression level of *oc-rtk* is knocked down by 25.8% in *oc-rtk*:RNAi nematodes. (**D**–**I**) Longevity and age-related changes of physiological processes of *O. chongmingensis* is reduced significantly in *oc-rtk*:RNAi nematodes, while those in the *oc-akt-1*:RNAi nematodes or in the *oc-pdk-1*:RNAi nematodes improved significantly. (**J**–**L**) Heat resistance in *oc-akt-1*:RNAi and *oc-pdk-1*:RNAi nematodes is improved, and so is the antioxidant activity in *pdk-1* RNAi nematodes. * *p* < 0.05; ** *p* < 0.01; *** *p* < 0.001.

**Figure 4 ijms-23-14692-f004:**
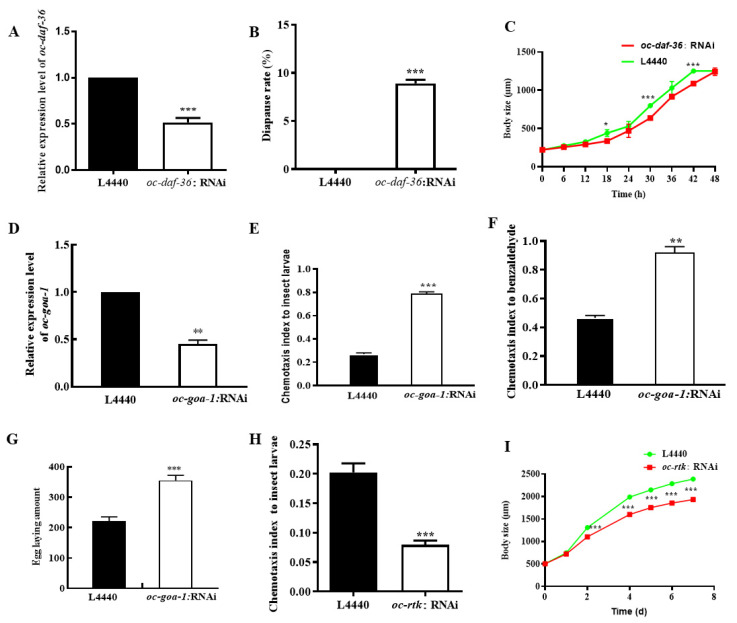
Identification of functions of *oc-daf-36, oc-goa-1*, and *oc-rtk* in regulating chemotaxis and fecundity of *O. chongmingensis*. (**A**) The expression level of *oc-daf-36* is knocked down by 48.3% in *oc-daf-36*:RNAi nematodes. (**B**) The diapause rate of IJs of *oc-daf-36*:RNAi nematodes is 8.9%. (**C**) Adults of *oc-daf-36*:RNAi nematodes need four hours more to reach the largest length. (**D**) The expression level of *oc-goa-1* is knocked down by 55.2% in *oc-goa-1*:RNAi nematodes. (**E**) Increased chemotaxis to wax worms in *oc-goa-1:*RNAi *O. chongmingensis*. (**F**) Increased chemotaxis to benzaldehyde in *oc-goa-1:*RNAi *O. chongmingensis*. (**G**) Increased number of eggs laid in *goa-1*:RNAi *O. chongmingensis* was increased. (**H**) Decreased chemotaxis to wax worms in *oc-rtk:*RNAi *O. chongmingensis*. (**I**) Decreased body length in *oc-rtk:*RNAi *O. chongmingensis*. * *p* < 0.05; ** *p* < 0.01; *** *p* < 0.001.

**Figure 5 ijms-23-14692-f005:**
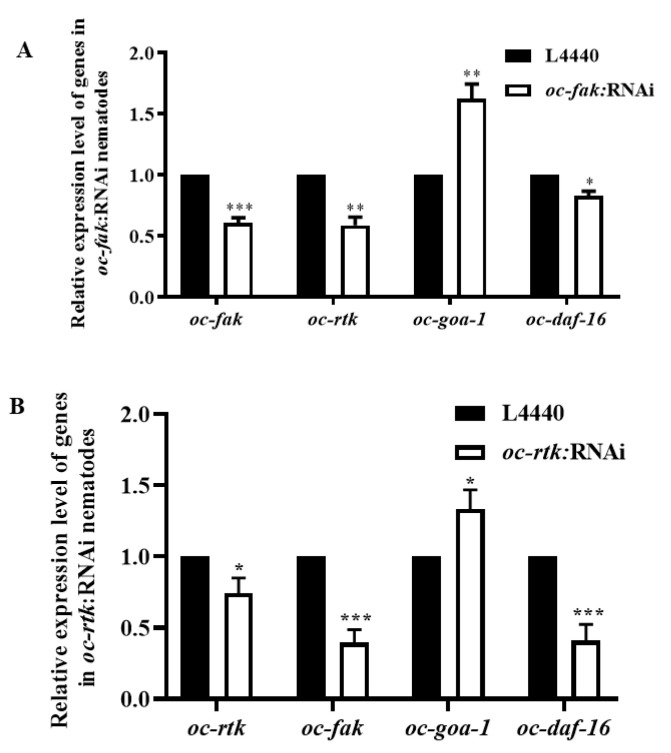
Interaction among the two tyrosine kinases *oc-fak and oc-rtk* at the transcriptional level, and the influence of them on the expression of *oc*-*daf-16* and *oc-goa-1.* (**A**) Upregulated expression of *oc-goa-1* and downregulated expression of *oc*-*daf-16* and *oc-rtk* in *oc-fak*:RNAi *O. chongmingensis*. (**B**) Upregulated expression of *oc-goa-1* and downregulated expression of *oc*-*daf-16* and *oc-fak* in *oc-rtk*:RNAi *O. chongmingensis*. * *p* < 0.05; ** *p* < 0.01; *** *p* < 0.001.

**Figure 6 ijms-23-14692-f006:**
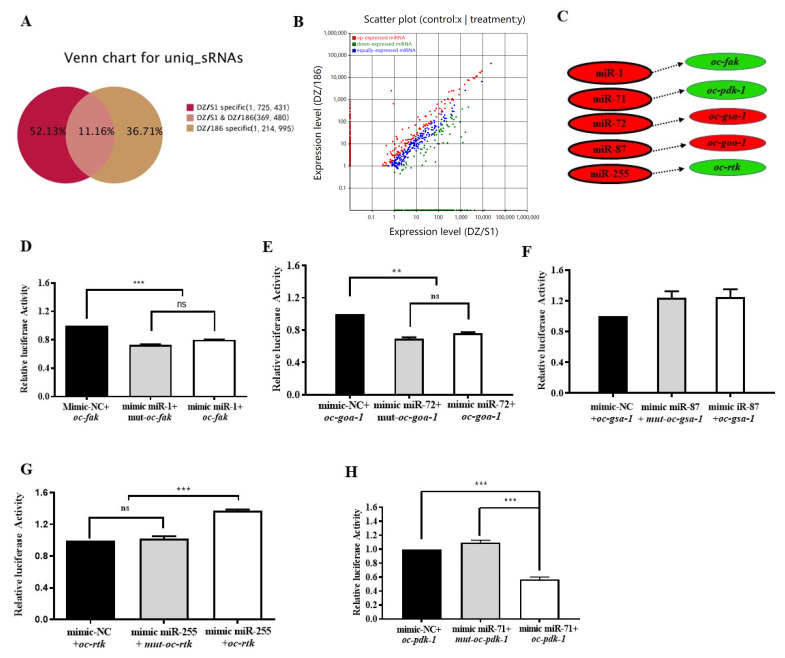
Global comparison analysis of miRNA sequencing data of native vs. non-native monoxenic *O. chongmingensis* and luciferase activities of related 3′ UTR reporter genes. (**A**) Venn chart analysis for uniq-sRNAs of DZ/186 and DZ/S1. (**B**) Analysis of differentially expressed miRNAs between DZ/186 and DZ/S1. (**C**) Prediction model of miRNAs interacting with mRNAs of the five selected genes. Luciferase activities of the *oc-fak* 3′ UTR reporter (**D**), the *oc-goa-1* 3′ UTR reporter (**E**), and the *oc-gsa-1* 3′ UTR reporter (**F**) have no significant difference, respectively, compared with both the miRNA control and the 3′ UTR mutant. (**G**) The luciferase activity of the *oc-rtk* 3′ UTR reporter is significantly increased compared with both the miRNA control and the 3′ UTR mutant. (**H**) The luciferase activity of the *oc-pak-1* 3′ UTR reporter is significantly decreased compared with both the miRNA control and the 3′ UTR mutant. ** *p* < 0.01; *** *p* < 0.001; ns, no significant difference.

**Figure 7 ijms-23-14692-f007:**
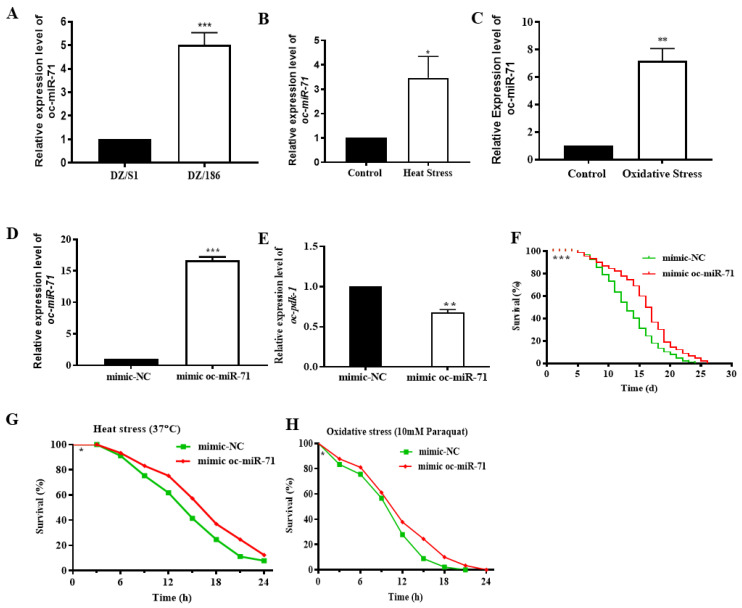
The effects of oc*-*miR-71 and its target *oc-pdk-1* on longevity and stress resistance of *O. chongmingensis*. (**A**) The expression of oc*-*miR-71 was upregulated four-fold more in DZ/186 compared with DZ/S1, as revealed by RT-qPCR. The expression of oc*-*miR-71 was upregulated two-fold more in *O. chongmingensis* treated with heat stress (37 °C) (**B**) or six-fold more with 10 mM paraquat (**C**) for 5 h. (**D**) The expression of oc*-*miR-71 was over-expressed 15-fold after soaking *O. chongmingensis* with mimic oc*-*miR-71. (**E**) The expression of *pdk-1* in *O. chongmingensis* was downregulated by 22.3% after soaking with oc*-*miR-71. (**F**–**H**) Longevity, heat resistance, and antioxidant activity were improved in *O. chongmingensis* with soaking with mimic oc-miR-71. * *p* < 0.05; ** *p* < 0.01; *** *p* < 0.001.

**Figure 8 ijms-23-14692-f008:**
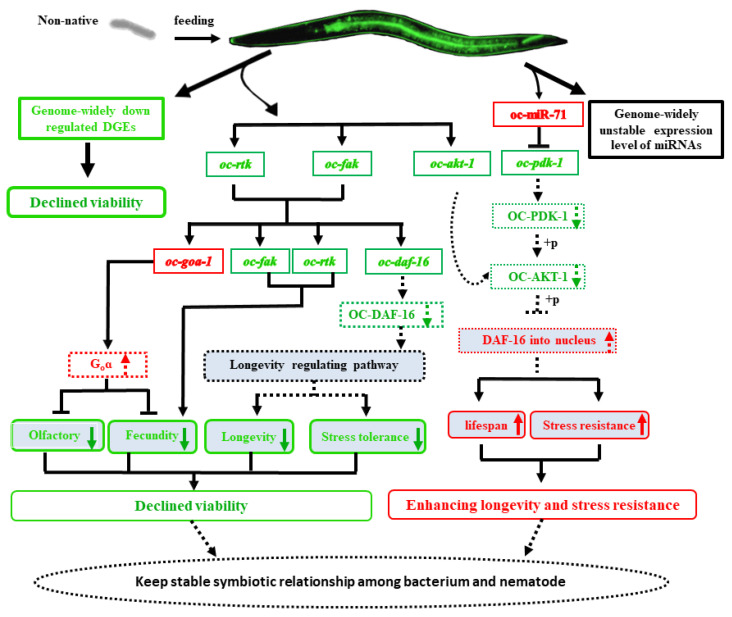
Predicted interaction network between *Oscheius chongmingensis* and its intestinal bacterial partner at global transcriptional and post-transcriptional levels. The single intestinal bacterial species *S. nematodiphila* orchestrates its nematode host’s viability and the stability of their symbiotic relationship at the global transcriptional and post-transcriptional levels, in part by regulating *oc*-*rtk*, *oc-fak*, *oc*-*akt-1*, and oc-miR-71. *Oc-fak* and *oc-rtk* play key roles in the viability of regulatory mechanisms of *O. chongmingensis* by positively mediating the expression levels of each other and *oc-daf-16* to indirectly impact stress tolerance and longevity, and by negatively mediating that of *oc-goa-1* to indirectly influence fecundity and olfaction of the nematode. A response mechanism model at transcriptional and post-transcriptional levels for enhanced survival of *O. chongmingensis* when faced with the non-native intestinal bacterial strain 186 was also deduced. For a response at the transcriptional level, expression of *oc-akt-1* is repressed in the non-native combination, which results in longer longevity and stronger stress tolerance to high temperature and oxidative stress through the insulin-like signaling (ILS) pathway. In the post-transcriptional level response, an upregulated miRNA, oc*-*miR-71, downregulates the target gene *oc-pdk-1* in the ILS pathway, improving the longevity and stress tolerance of *O. chongmingensis*. Genes with downregulated expression in the non-native combination are indicated in green boxes and marked with green color, while genes with upregulated expression in the non-native combination are indicated in red boxes and marked with red color. Solid boxes indicate verified in this study. Dashed boxes indicate induced according to previously verified facts.

**Table 1 ijms-23-14692-t001:** Effects of *Serratia nematodiphila* strains on development and reproduction of the first generation IJs of *Heterorhabditidoides* in different combinations.

Combination	Developmental Rate (%)	Number of Second Generation IJs per Insect Larva	IJs Size	
			Length (μm)	Max Width (μm)
DZ/S1	85.43	6782	425.7 ± 25.43	22.6 ± 3.1
DZ/186	26.72	386	398.52 ± 35.25	20.3 ± 4.6
RG/S1	75.19	5873	569.4 ± 36.72	27.38 ± 3.61
RG/186	83.52	4950	582.7 ± 38.32	28.62 ± 2.65

The positive correlation relationship index between female percentage and IJs yield r = 0.971; *p* < 0.05.

**Table 2 ijms-23-14692-t002:** The lifespan of *H. chongmingensis* (DZ) worms cultured in NGM liquid medium with cognate bacterial strain *S. nematodiphila* S1 and non-cognate bacterial strain *S. nematodiphila* 186.

Groups	No. of Worms	Mean Lifespan (Days)	Max Lifespan (Days)	Median Lifespan	Log-Rank Test
DZ/S1	90	13.10	24	14	*p = 0.0005*
DZ/186	90	11.28	22	10	*p = 0.0005*

## Data Availability

DGE and sRNA library sequencing data are available from the NCBI BioProject database, accessions #PRJNA565717 and PRJNA566281. All other relevant data are included within the manuscript and its Appendix A files.

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
