# Peer review of "An Intestinal Symbiotic Bacterial Strain of Oscheius chongmingensis Modulates Host Viability at Both Global and Post-Transcriptional Levels"

_ijms, 2022, doi:10.3390/ijms232314692_

Round 1

Reviewer 1 Report

The author report in their intriduction that O. chongmingensis is a 'putative' entomopathogenic nematode, howevre  in the rest of the mansurcipt they consider the nematide as an entomopathogenic nematode. This seems to be controversial. In my opinion Oscheius chongmingensis is should not be considered an entomopathogenic nematode. This nematode can survive  with any other bacterium including E. coli OP 50.

 The authors state "thee intestinal microbiome plays a crucial role in mediating diverse host biological  processes, including host–symbiont interactions" But to  consider in  this study  the intreactions with one single bacterium is streching the concept of microbiome too much...

In te materials and methods section the authors indicate they used 10 wax worms in onePetri dish to set up infections.  This procedure is not suitable to assess  viriulence because the inoculum to a dish will not allow to determine the number of IJs that infect a single insect and thus the number of IJs penertating or infecting each insect is variable.  The proper experiment should comsider a 1: 1 assay. THis also applyes to the rest of the wetlab experiments

The authors disregard recent literature related to a ranscriptomic analysis in Steinernema nematodes by Lefoulon elt al. 2022 and other related work on Steinernema-Xenorhabdus fitness by multiple authors including but not limkited to McMullen 2015, Roder et al., 2018, 2019, etc)

Reviewer 2 Report

The present paper could be well for the symbiosis biology.  However, the part "Materials and methods" MAY NOT be ahead of the part "Result and discussion".  Hence, it was NOT easy to read and think the contents, unfortunately.   If the authors DO NOT intend to change the their own style, you (responsible editor of this scientific journal or someone like that) and that guys had better talk about this matter.  

Author Response

Yes. We put the part "Result and discussion" ahead of "Materials and methods"  in our manuscript for reading easy going.